# Tomographic reconstruction algorithms for retrieving two-dimensional ice cloud microphysical parameters using along-track (sub)millimeter-wave radiometer observations

Yuli Liu[1,2] and Ian S. Adams[2]

[1]Goddard Earth Sciences Technology and Research (GESTAR) II, University of Maryland Baltimore County (UMBC), Baltimore, MD 21250, USA
[2]NASA Goddard Space Flight Center, Greenbelt, MD 20771, USA

**Correspondence:** Yuli Liu (yuliliu@umbc.edu)

**Abstract.** The submillimeter-wave radiometer operating in the along-track scanning mode continuously collects brightness temperature (TB) data over a two-dimensional (2D) cloud cross-section as the platform moves forward. TB observations from multiple positions and viewing angles show great promise in better constraining the 2D cloud microphysical properties compared to single-angle observations. In this study, we develop two types of tomographic reconstruction algorithms to retrieve 2D ice water content (IWC) profiles using multi-angle TB observations. The one-dimensional (1D) tomographic algorithm performs 1D retrievals beam by beam using each TB observation at a specific sensor position and viewing angle to derive cloud properties along the propagation path. It then integrates the 1D retrieval results to construct 2D cloud cross-section. The 2D tomographic algorithm directly constrains the 2D cloud microphysical properties using multi-angle scanning TB observations. Starting with an initial assumption, the algorithm iteratively refines the 2D cloud microphysical quantities by minimizing discrepancies between TB simulations and observations under prior constraints. Both tomographic algorithms are developed based on a hybrid of Bayesian Monte Carlo Integration (MCI) and Optimal Estimation Method (OEM). A simulation experiment is conducted to evaluate the performance of two tomographic reconstruction algorithms. The experiment demonstrates stable convergence of both tomographic methods, with the 2D tomographic algorithm exhibiting superior performance. The experiment results highlight the significant advantage of using multi-angle observations to constrain 2D cloud structure. Compared to nadir-only retrievals, the tomographic technique provides a detailed reconstruction of ice clouds' inner structure with high spatial resolution. Also, the technique significantly improves retrieval accuracy by correcting systematic biases and reducing the derivation of retrieval errors. Furthermore, the tomography technique effectively increases detection sensitivity for small ice cloud particles.

## 1 Introduction

Submillimeter-wave radiometry is a promising technique for studying ice clouds in the atmosphere (Evans et al., 1999; Buehler et al., 2007). The wavelengths in the submillimeter range are comparable to the size of ice cloud particles, making the scattering interactions between the electromagnetic waves and ice particles significant. Unlike visible and infrared wavelengths that are

often absorbed and scattered by cloud tops, submillimeter waves can penetrate deeper into cloud layers and provide more information about the internal structure of clouds. Submillimeter channels at different frequencies are sensitive to different parts of the ice particle size distribution. With carefully selected channels, the multi-frequency observations enable more precise retrievals of cloud properties, such as ice mass and particle size (Buehler et al., 2012). Field campaigns like CRYSTAL–FACE (Cirrus Regional Study of Tropical Anvils and Cirrus Layers – Florida Area Cirrus Experiment (Evans et al., 2005)), TC4 (Tropical Composition, Cloud and Climate Coupling (Evans et al., 2012)), and IMPACTS (Investigation of Microphysics and Precipitation for Atlantic Coast-Threatening Snowstorms (McMurdie et al., 2022)) have demonstrated the effectiveness of submillimeter radiometry in airborne measurements. Looking forward, the deployment of new satellite sensors, such as POLSIR (Polarized Submillimeter Ice-cloud Radiometer), ICI (Ice Cloud Imager (Eriksson et al., 2020)), and the radiometer on the AOS (Atmosphere Observing System (Braun et al., 2022)) constellation, will further extend the capability to study ice clouds on a global scale (Wu et al., 2024).

Submillimeter-wave radiometer measurements are inherently sensitive to column-integrated quantities like the ice water path (IWP). That is because the observed brightness temperature (TB) captures the cumulative scattering and absorption effects along the sensor's entire line of sight through the atmosphere. Several algorithms have been developed to retrieve the IWP from submillimeter-wave TB measurements. Two notable approaches are Bayesian Monte Carlo Integration (MCI) (Evans et al., 2002; Eriksson et al., 2020) and Neural Network (NN) (Brath et al., 2018; Pfreundschuh et al., 2018). The Bayesian MCI uses a stochastic approach and relies on a pre-established a priori database of atmospheric and cloud states to estimate the posterior probability distribution given an observed TB. For a NN trained by a priori database, the algorithm efficiently retrieves cloud parameters by recognizing patterns and correlations within the data. Both BMCI and NN algorithms have been rigorously evaluated using simulated data and real observations from airborne campaigns. Given that TB measurements are sufficiently sensitive to IWP, both algorithms have demonstrated good retrieval performance and are well-prepared for operational deployment in retrieving IWP when space-borne observations are available.

Some algorithms have been developed to explore the feasibility of retrieving ice water content (IWC) profiles from single-view TB observations. In addition to database-driven approaches like Bayesian MCI and NN (Wang et al., 2017), optimization techniques such as the Optimal Estimation Method (OEM) (Evans et al., 2012; Pfreundschuh et al., 2020) and the Ensemble Optimization Method (Liu et al., 2018; Liu and Mace, 2022) have been designed to refine cloud microphysical properties by better aligning observations with forward model simulations. These studies indicate that single-view TB observations do exhibit some sensitivity to localized variations in ice particles, particularly in regions with high IWC. However, this sensitivity is limited, and the retrieval results of IWC profiles are not satisfactory as the retrieved cloud quantities lack accuracy and the uncertainty estimates are significant. Additionally, a substantial portion of the constraints on the posterior probability density comes from a priori information rather than the observations themselves. How to construct ice cloud vertical profiles using passive TB observations remains a daunting challenge.

In an along-track scanning mode, the (sub)millimeter-wave radiometer continuously scans a two-dimensional (2D) cross-section of the cloud in one direction beneath the spacecraft or aircraft platform as it moves forward. Each TB beam captures radiation that penetrates the atmosphere from a unique propagation path. Integrating multi-view observations from different

positions and viewing angles shows great potential to more effectively constrain the 2D ice cloud structure. Reconstruction techniques using multi-angle data have long been used in Computed Tomography (CT) within the medical field. In cloud remote sensing, several tomographic algorithms have been developed for passive optical instruments (Levis et al., 2020; Forster et al., 2021; Loveridge et al., 2023), and some studies have investigated the application of microwave tomography methods to liquid clouds (Warner et al., 1985; Huang et al., 2008). However, to the best of our knowledge, no tomographic cloud reconstruction algorithms have yet been developed in the (sub)millimeter-wave radiometry field.

In this paper, we develop two types of tomographic retrieval algorithms to reconstruct the inner structure of ice clouds using along-track scanning (sub)millimeter-wave TB observations. The objective is to leverage the multi-view observations to improve the retrieval accuracies of the 2D IWC cloud structure. We conduct a simulation experiment to evaluate the retrieval performance of both tomographic reconstruction algorithms. The paper is structured as follows: Section 2 introduces the passive radiometer sensor, the radiative transfer model, and the simulated multi-angle TB observations. Section 3 describes the pre-established retrieval database used in the reconstruction algorithms. Section 4 presents the tomographic reconstruction algorithms in detail, followed by Section 5, which discusses the evaluation results of these algorithms. Finally, Section 6 presents the summary and conclusions.

## 2 Simulated observations

### 2.1 CoSSIR radiometer

This study assumes a radiometer based on the Configurable Scanning Submillimeter-wave Instrument/Radiometer (CoSSIR, formerly the Compact Scanning Submillimeter Imaging Radiometer (Evans et al., 2005)). Along with ISMAR (International Submillimeter Airborne Radiometer, (Fox et al., 2017)), CoSSIR is one of two operational airborne submillimeter-wave radiometers. CoSSIR has a long history of deployment in many field campaigns, such as CRYSTAL-FACE and TC4. CoSSIR has recently been completely updated with new receives. The updated CoSSIR channels cover a frequency range from 170 GHz to 684 GHz, and all channels have dual-polarization capabilities. The central frequencies and associated Noise Equivalent Differential Temperatures (NeDT) are summarized in Table 1. The updated CoSSIR was first deployed during the final phase of the IMPACTS field campaign in 2023, where the sensor operates in a hybrid conical and along-track scanning mode.

In the simulation experiment for this study, CoSSIR is configured to operate in a hypothetical along-track scanning mode, with scanning occurring in a single direction. CoSSIR is assumed to be mounted on a high-altitude Earth Resource 2 (ER-2) aircraft and fly over a designated reference cloud scene. A segment of CloudSat orbit 127-021225 in 2008 is selected as the reference cloud scene, and the 2D cross-section of IWC is obtained from the 2C-ICE product (Deng et al., 2015). The top panel of Figure 1 shows the selected IWC curtain and the bottom panel shows the corresponding IWP. Compared to numerical weather models that involve many assumptions and simplifications, the CloudSat dataset offers more realistic distributions of ice cloud microphysical quantities. The selected cloud segment covers a latitudinal range from $44°$ to $52°$, and it contains a total of 842 vertical cloud columns. The IWC cross-section exhibits varying patterns and the IWP fluctuates between 0.1 $kg\ m^{-2}$ to 5 $kg\ m^{-2}$, providing sufficient cloud variations to evaluate the tomographic reconstruction algorithms. The liquid cloud

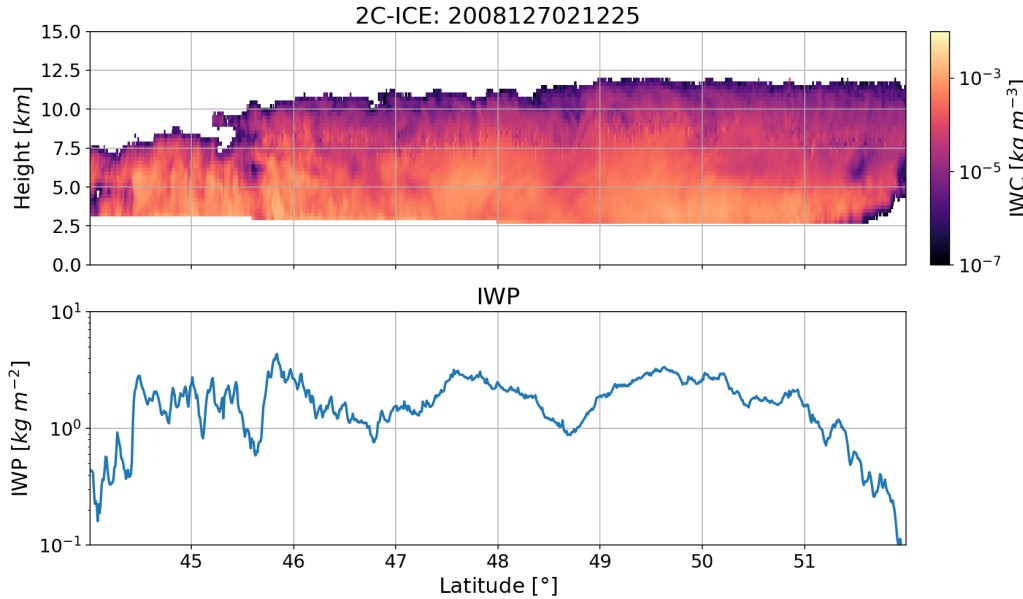

**Figure 1.** The top panel shows the reference ice water content (IWC) profiles that are obtained from the CloudSat 2C-ICE product for the 127021225 CloudSat orbit in 2008. The bottom panel shows the corresponding ice water path (IWP).

profiles are obtained from the corresponding 2B-CWCRO product (Austin et al., 2009) and other atmospheric and surface parameters are obtained from the ECMWF dataset.

The imaginary ER-2 aircraft equipped with the CoSSIR sensor is assumed to follow the same south-to-north trajectory as the CloudSat orbit but at an altitude of 20 km. The ER-2's ground speed is set to 208 meters per second. The scanning parameters
for the along-track CoSSIR are aligned with those used in the actual CoSSIR deployment during the IMPACTS campaign: the sector width is set to 98°, the scanning speed to 100° per second, and the integration time to 10 milliseconds. Based on the scanning parameters, each scan slice sweeping from one side to the other contains 97 beam observations. During the flight over the 875-kilometer reference cloud, CoSSIR completes a total of 1173 slices, resulting in 113,781 beam observations overall.

## 2.2 Radiative transfer model

A radiative transfer model based on ARTS (Buehler et al., 2018) is developed to simulate the along-track scanning TB observations for CoSSIR. The ray-tracing method with Independent Beam Approximation (IBA) is used to simplify the three-dimensional (3D) simulation by approximating it as a set of one-dimensional (1D) simulation setups. The IBA treats the radiative transfer calculation along a specific line of sight independently of the others. While the IBA simplification neglects the multi-scattering effects caused by spatial variations in ice cloud properties, it is still the most practical method available
for the along-track scanning simulation as complete 3D radiative transfer models are too computationally expensive for the

**Table 1.** Channel characteristics of the CoSSIR radiometer.

| Frequencies (GHz) | Polarization | NeDT(K) |
|:---:|:---:|:---:|
| 170.5 | V/H | 0.2 |
| 177.31 | V/H | 0.2 |
| 180.31 | V/H | 0.2 |
| 182.31 | V/H | 0.2 |
| 325.15±11.5 | V/H | 1.5 |
| 325.15±3.4 | V/H | 1.5 |
| 325.15±0.9 | V/H | 1.5 |
| 684.0 | V/H | 1.0 |

purposes of this study. Each beam is assumed to be a pencil beam of radiation, and the effects of beam width and footprint is deferred to future studies.

In the ARTS forward model, the ER-2 aircraft is positioned within a 3D atmospheric coordinate system. Instead of establishing a 2D coordinate framework, a 3D framework enables the future extension to conical and cross-track scanning modes, as well as to real-world observations that requires consideration of factors like beam width and aircraft rotation. The horizontal resolution of the coordinate system is specified at 1 kilometer to align with the CloudSat dataset. The vertical resolution is set to 250 meters. To simulate the sensor's observation at a specific position and viewing angle, a pencil-beam traces the path of radiation between the sensor and the surface. The ray-tracing method with IBA records the geodetic coordinates (latitude, longitude, and altitude) of the voxels along the line of sight. The atmospheric and cloud parameters along the radiation's propagation path are extracted, and a 1D radiative transfer model is employed to compute the TB simulation for this beam. The geodetic coordinates of the voxels along each line of sight are also utilized in the cloud reconstruction algorithms discussed in Section 4.

In the 1D ARTS radiative transfer model, the absorbing gases include $O_2$, $N_2$, $O_3$, and $H_2O$. The forward model contains two types of scattering cloud hydrometeors including ice clouds and liquid clouds. Both cloud particle types are assumed to be randomly oriented, and their scattering properties are obtained from the ARTS single scattering database (Eriksson et al., 2018). The plate aggregate habit is chosen to characterize the ice cloud particles, and the liquid cloud particles are modeled as spheres. The midlatitude Field07 scheme (Field et al., 2007) is used to parameterize the particle size distribution (PSD) for ice clouds, and the Mono-dispersive PSD scheme is used for liquid clouds. For the ocean surface, emissivity is calculated using the TESSEM (Tool to Estimate Sea-Surface Emissivity from Microwave to Submillimeter Waves (Prigent et al., 2017)) model. For the land surface, emissivity is determined using a dataset developed from GPM measurements (Munchak et al., 2020). The DISORT (DIScrete-Ordinate-method Radiative Transfer (Stamnes et al., 1988)) is used as the scattering solver, and the simulations are currently limited to the first Stokes parameter $I$. The utilization of polarized TB information will be deferred to future studies.

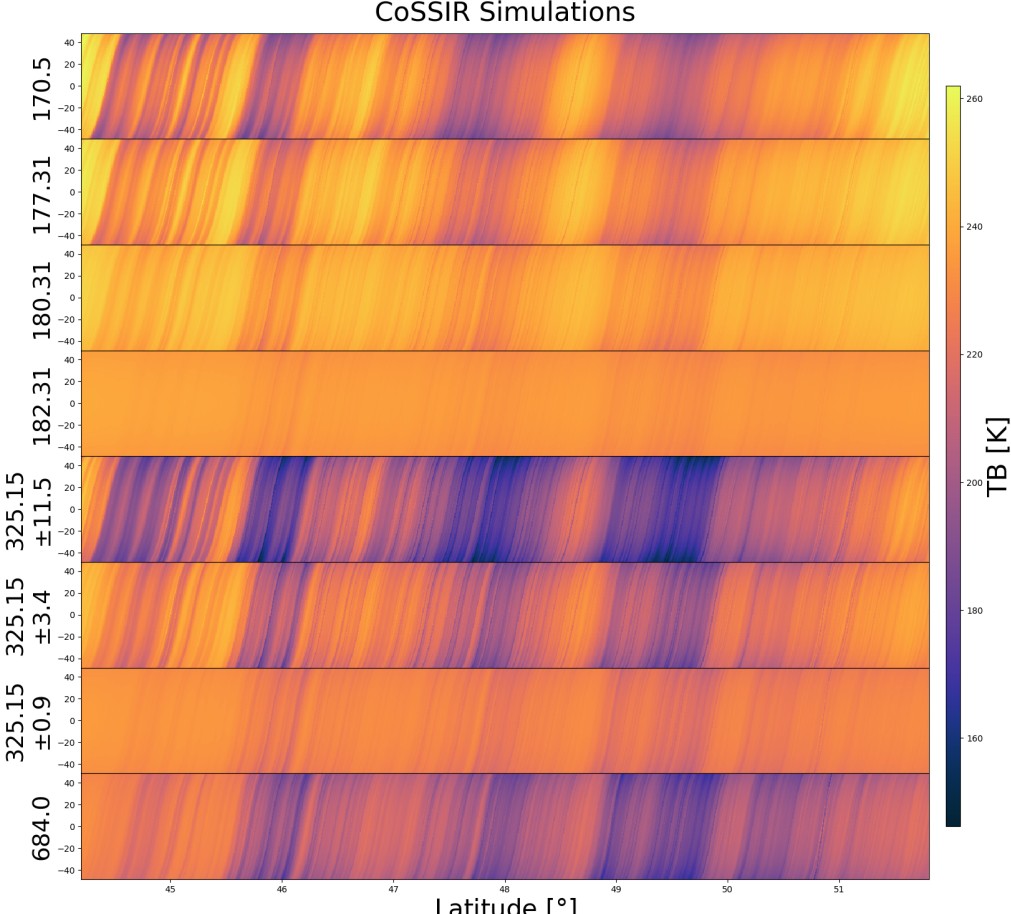

**Figure 2.** The simulated along-track scanning brightness temperature (TB) observations as a function of latitude and scanning angles for different CoSSIR channels as the ER-2 aircraft flies over the reference cloud scene.

Figure 2 shows the along-track scanning TB simulations as a function of latitude and scanning angles for different CoSSIR
channels as the ER-2 airplane moves forward. As mentioned in Section 2.1, the along-track scanning observations over the selected trajectory consist of 1,173 slices, with each slice containing 97 beams. As the radiometer continuously scans in a single direction, distinct inclined patterns are observed for each channel as the airplane moves ahead. The 182.31 GHz and 325±0.9 GHz channels show the lowest sensitivities to variations in ice cloud parameters due to water vapor absorptions at and above the cloud layers. The 325±11.5 GHz and the 684 GHz channels exhibit the deepest TB depressions when encountering
deep clouds.

Figure 3 shows the simulated TB observations at nadir for the CoSSIR channels over the reference cloud scene. The nadir-view TB simulations will also be input into the retrieval algorithm presented in Section 4 to retrieve vertical profiles of IWC. The intention is to evaluate the improvements achieved by using multi-angle observations compared to single-view observa-

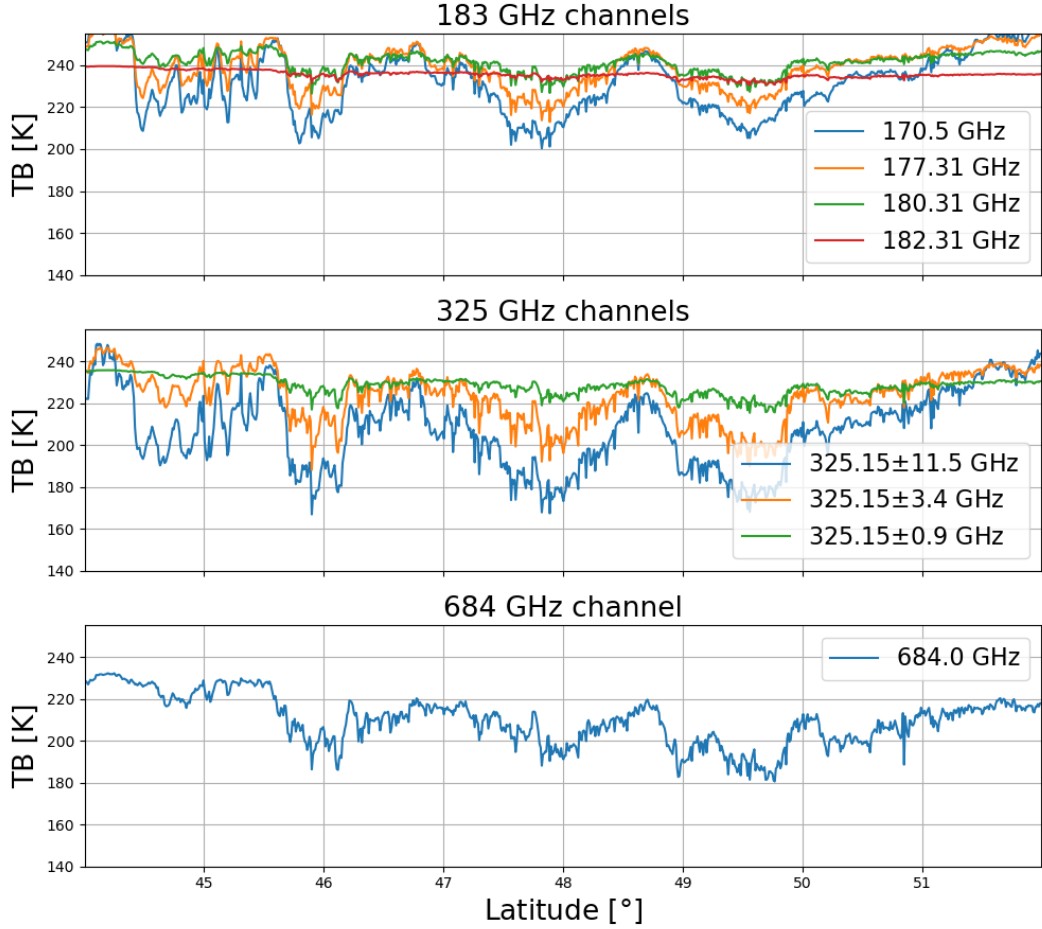

**Figure 3.** Same as Figure 2 but only for the simulated observations at the nadir viewing angle.

tions. The figure clearly shows the varying sensitivities of ice cloud parameters at different channels, and no channels show
signs of saturation at the selected IWP range.

## 3 Retrieval database

As will be discussed in Section 4, the developed tomographic cloud retrieval algorithms are within the framework of Bayes' the-
orem and rely on an established multi-angle retrieval database to introduce the a priori constraint. A retrieval database consists
of two elements: the atmospheric and cloud parameters distributed according to our prior knowledge, and the corresponding
multi-angle TB simulations computed using a radiative transfer model.

The a priori cloud profiles are also obtained from CloudSat products. Atmospheric and cloud columns from the northern
midlatitudes (30°N to 60°N) at a similar time as the reference clouds but in a different year are selected as the a priori profiles.

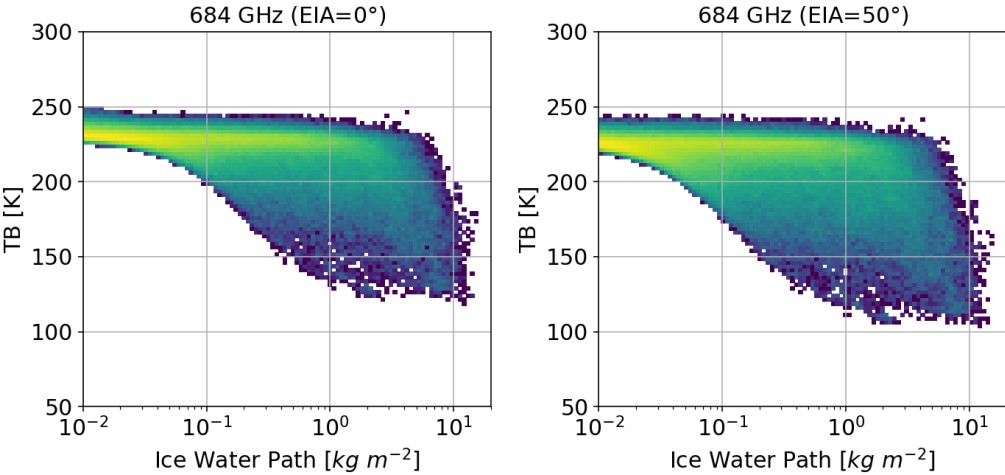

**Figure 4.** The TB simulations at 684 GHz for the a priori atmospheric and cloud profiles at nadir and the $50°$ incidence angle, respectively. (EIA: Earth Incidence Angle)

Columns from March, April, and May in 2007 that are within the latitudinal range and with an IWP greater than $10^{-2}\ kg\ m^{-2}$ are selected. Approximately 500,000 hydrometer profiles meeting these criteria are chosen. Similarly, the ice cloud profiles, liquid cloud profiles, and other atmospheric and surface parameters are obtained from the 2C-ICE, 2B-CWCRO, and ECMWF datasets, respectively.

The same 1D ARTS forward model presented in Section 2.2 is employed to compute the TB simulations. For each a priori atmospheric and cloud profile, multiple scanning angles ranging from nadir to 50° with 1° increments are computed for the CoSSIR channel frequencies to build up a multi-angle retrieval database. It should be noted that the current database performs multiple-angle TB simulations using the same vertical cloud profile, and the actual horizontal cloud variations at different layers in the CloudSat products are not captured. This simplification reduces the variability of cloud parameters in the prior database and affects the correlations between cloud parameters at different layers. The establishment of a more sophisticated prior database is deferred to future work. Figure 4 shows the TB simulation at 684 GHz as a function of IWP for both nadir and 50° viewing angle. The IWP ranges from $10^{-2}\ kg\ m^{-2}$ to over $10\ kg\ m^{-2}$, with the 684 GHz TB simulations dropping as low as 130 K at the nadir. The database densely covers the range of IWP variation in reference clouds shown in Figure 1. The TB at a 50° viewing angle shows greater depressions for the same cloud profile due to the inclined line of sight.

## 4 Tomographic cloud reconstruction algorithms

We develop two types of tomographic cloud retrieval algorithms to reconstruct the internal structure of the 2D cloud cross-section using multi-view observations. Both algorithms are based on the Bayesian theorem, which integrates a priori knowledge

with the observed data to provide a probabilistic estimate of cloud parameters. The first tomography algorithm applies the 1D retrieval algorithm to each TB beam to determine the cloud properties along the propagation path, and then combines the 1D retrieved parameters to construct the 2D cloud structure. The second tomographic algorithm directly constrains the 2D cloud microphysical properties using multi-angle scanning TB observations. Starting with an initial assumption of the 2D cloud quantities, the algorithm iteratively compares simulations with the actual scanning observations and updates the cloud properties under appropriate a priori constraints. Both tomographic reconstruction algorithms rely on a hybrid Bayesian MCI and OEM methodology. The MCI uses the pre-established retrieval database to derive the initial assumption and the a priori constraint for the optimization process, and the OEM further refines the cloud parameters to maximize the posterior probability density function (PDF).

## 4.1    1D tomographic reconstruction algorithm

Figure 5 shows the flowchart of the 1D tomographic reconstruction algorithm. The fundamental idea is to perform the 1D retrieval beam by beam and then combine the retrieved 1D cloud parameters along each straight ray path to construct the 2D cloud structure. The geodetic coordinates of the voxels interacting with independent pencil beams that are obtained through the ray-tracing technique discussed in Section 2.2 are used to guide the cloud construction process. The 1D hybrid Bayesian MCI and OEM algorithm is employed to retrieve cloud parameters for each TB beam. The 1D retrieval process begins by applying the Bayesian MCI using the multi-angle retrieval database established in Section 3. If the MCI fails due to an insufficient number of database cases within the measurement uncertainties, the OEM optimization procedure is employed to further maximize the posterior PDF. In the 1D OEM algorithm, the measurement vector consists of the pencil-beam TB observation at a specific sensor position and viewing angle, and the state vector consists of the IWC profile along a corresponding propagation path.

The Bayesian MCI algorithm uses Monte Carlo sampling to approximate the posterior distribution of the cloud parameters. The atmosphere/cloud profiles in the retrieval database are distributed according to the a priori PDF. The posterior PDF is estimated by calculating the conditional PDF measuring the likelihood of the observation given a particular atmospheric state:

$$P_{post}(x|y_{obs}) \propto exp(-\frac{1}{2}\chi_y^2) \qquad \chi^2 = \sum_{j=1}^{N} \frac{(y_{sim,j} - y_{obs,j})^2}{\sigma_j^2} \tag{1}$$

where $y_{sim,j}$ and $y_{obs,j}$ are the simulated and observed TB in the $j$th channel, the $\sigma_j$ is the measurement noise, and $N$ is the number of radiometer channels. Retrieval results and uncertainty estimates are derived by integrating over the possible states that fit the input observations:

$$x_{ret} = \sum_i x_i * p_{post}(x_i|y_{obs}) \tag{2}$$

$$\sigma_x^2 = \sum_i (x_i - x_{ret})^2 * p_{post}(x_i|y_{obs}) \tag{3}$$

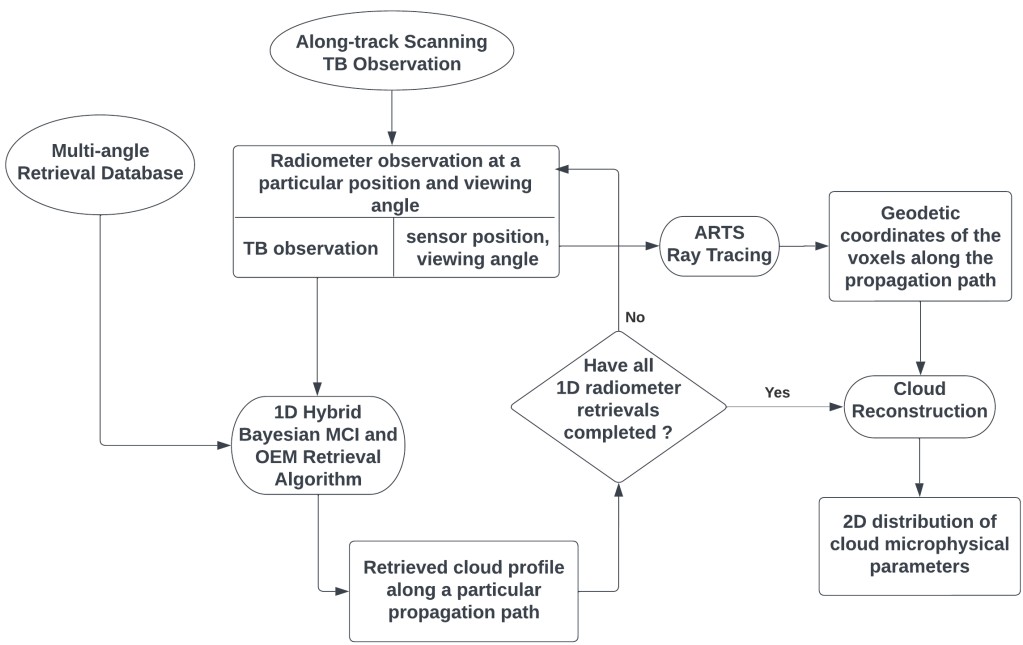

**Figure 5.** Flowchart of the one-dimensional (1D) tomographic cloud reconstruction algorithm.

The covariance between different states is also computed by summing over the weighted cases:

$$cov(m.n) = \sum_i (x_{i,m} - x_{ret,m}) * (x_{i,n} - x_{ret,n}) * P_{post}(x_i|y_{obs}) \tag{4}$$

where $m$, $n$ represent the $m$th and $n$th states in the state vector. A successful implementation of the Bayesian MCI requires a minimum of database cases (25 cases in this study) to be found within a specified $\chi_y^2$ threshold (Evans et al., 2005). The $\chi_y^2$ is specified as $m + 4\sqrt{m}$, where $m$ is the number of radiometer channels. If this criterion is not met, the Bayesian MCI algorithm is considered to have failed, and the OEM optimization process starts to further refine the cloud parameter estimates.

The OEM maximizes the posterior PDF by minimizing the following cost function:

$$\chi^2 = (y_{sim} - y_{obs})^T S_y^{-1}(y_{sim} - y_{obs}) + (x - x_a)^T S_a^{-1}(x - x_a) \tag{5}$$

where $y_{sim}$ and $y_{obs}$ represent the simulation and observation vector, $S_y$ is the measurement error covariance, $x_a$ is the a priori value of the state vector, and $S_a$ is the associated covariance matrix. The OEM algorithm requires the prior PDF to be in Gaussian form. Considering that the global a priori PDF of IWC in the retrieval database established in Section 3 is highly non-Gaussian, the OEM relies on the posterior PDF estimated by the Bayesian MCI step to derive an appropriate local Gaussian a priori constraint (Liu et al., 2018, 2022). When fewer than the required number of database points are found within the $\chi_y^2$ threshold, the uncertainty of all radiometer channel $\sigma_j$ is inflated by a factor of $\sqrt{2}$ until the minimum number of points is reached (Evans et al., 2005; Liu and Mace, 2022). The $\sigma_j$ inflation step prevents the MCI from relying too heavily

on a small subset of database cases. Ensuring sufficient case diversity is important for producing reliable estimates of retrieval uncertainties. The posterior PDF is updated as:

$$p_{post}(x|y_{obs}) \propto exp(-\frac{1}{2m}\chi_y^2) \tag{6}$$

where $m$ is the inflation factor. The calculations of mean values, standard deviations, and the correlation coefficient in Eqs. (2-4) are updated accordingly.

In the 1D OEM algorithm, the a priori state $x_a$ is set as the IWC profile retrieved by Bayesian MCI after inflating measurement uncertainties. The diagonal elements of the covariance matrix $S_a$ are set as the variance of the retrieved state in Eq. (3), while the off-diagonal elements of $S_a$ are set as the covariance between different states computed in Eq. (4). Prior state $x_a$ is also used as the starting point for the OEM iteration process. Using the gradient information provided by the Jacobian matrix, the Levenberg-Marquardt (LM) minimization method (Rodgers, 2000) is implemented to iteratively search for the optimal

state. The Jacobian matrix is computed using the finite difference method, where perturbations are applied to each component of the state vector to determine the sensitivity of TB simulations to the cloud state. The OEM process stops when the step size becomes smaller than a specified threshold or the number of iterations exceeds a limit. The optimal estimation posterior error covariance of the state vector is computed to characterize the retrieval uncertainties:

$$S = (S_a^{-1} + K S_y^{-1} K)^{-1} \tag{7}$$

where $K$ is the Jacobian matrix.

The 1D hybrid Bayesian MCI and OEM algorithm is applied to each pencil-beam observation at different positions and viewing angles. Once this process is complete, the 1D retrieval results are integrated to construct the final representation of the 2D cloud microphysical parameters. Cloud properties at a specific grid point are constrained by multiple TB observations from different angles. Each TB beam yields a posterior PDF on the cloud quantities from the 1D hybrid Bayesian MCI and

230 OEM algorithm. The posterior PDF produced by the OEM optimization is Gaussian distributed. For the retrievals obtained using Bayesian MCI alone, it is reasonable to summarize the posterior PDF as Gaussian using the mean and standard deviation parameters in Eqs. (2-3) to facilitate the cloud construction analysis. In the final step, the cloud parameter at a specific voxel is determined by averaging all posterior PDFs imposed by the radiation beams that interact with this voxel. The averaged posterior PDF is also Gaussian distributed, and the mean and standard deviation are computed as follows:

$$x = \frac{1}{n}\sum_{k=1}^{n} x_k \qquad\qquad \sigma^2 = \frac{1}{n^2}\sum_{k=1}^{n} \sigma_k^2 \tag{8}$$

where $x_k$ and $\sigma_k^2$ refer to the retrieved state and associated variance from the $k$th TB observation, and $n$ is the number of the interacted TB observations at this grid point. The average method used here assumes that each TB observation from different angles contributes equally to the final results at a given voxel. The weighted averaged method based on information such as the covariance matrix of the posterior PDF and the $\chi^2$ values of the OEM results will be investigated in the future.

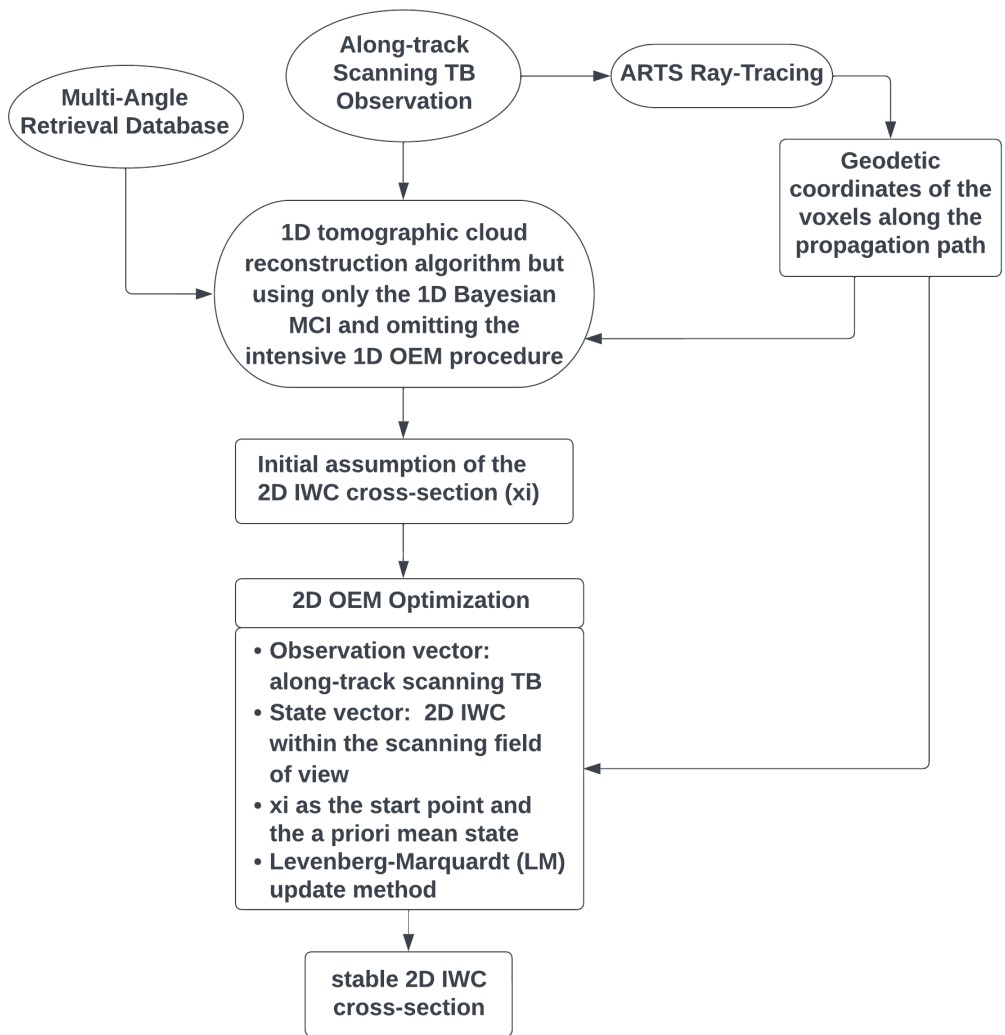

**Figure 6.** Flowchart of the two-dimensional (2D) tomographic cloud reconstruction algorithm.

## 4.2  2D tomographic reconstruction algorithm

Figure 6 shows the flowchart of the 2D tomographic reconstruction algorithm. The central concept is to use the continuously collected along-track scanning TB observations to simultaneously constrain the 2D cloud properties within the scanning field of view. The 2D tomography algorithm also employs the hybrid Bayesian MCI and OEM methodology. The Bayesian MCI algorithm is used to obtain the initial assumption of the 2D cloud structure. The OEM optimization is then applied to iteratively refine the 2D cloud microphysical parameters to minimize the discrepancies between simulated and observed TB values under a priori constraints.

The first step of the 2D tomographic cloud reconstruction algorithm is to obtain an initial estimate of the 2D cross-section of ice clouds. This is done by implementing the 1D tomographic reconstruction algorithm presented in Section 4.1 but using only the computationally efficient Bayesian MCI algorithm and omitting the intensive 1D OEM procedure. For a radiation beam at a specific position and viewing angle, the updated 1D Bayesian MCI algorithm that can automatically adjust the measurement uncertainty is applied to derive the retrieval results and uncertainties for the cloud parameters on the propagation path. The retrieved results at each grid point are averaged to create the posterior PDF estimates of the reconstructed IWC quantities. The mean and standard deviation of the reconstructed posterior PDFs from Eq. (8) are used to establish the subsequent 2D OEM optimization procedure.

In the 2D OEM optimization, the measurement vector consists of a series of along-track scanning TB observations, and the state vector consists of the 2D IWC quantities within the scanning field of view. The prior state vector $x_a$ is set as the mean state of the reconstructed 2D IWC quantities using the 1D Bayesian MCI results. The 2D OEM optimization employs a local Gaussian a priori constraint as well. The setup for the a priori covariance matrix $S_a$ consists of three parts. The diagonal elements of $S_a$ are set as the variance of the posterior PDFs for the reconstructed quantities at each voxel, as computed in Eq. (8). For the off-diagonal elements of $S_a$, if two voxels are connected by a radiation ray, their covariance is determined by averaging the covariance computed from all radiation beams where the two voxels lie on the same ray path. The covariance of two voxels on the same radiation beam is calculated using Eq. (4). For voxels that cannot be connected by any straight ray beam, the Bayesian MCI reconstructed results do not provide their correlation information. Instead, the correlation coefficient is computed by randomly sampling ice cloud profiles in the retrieval database that have the same relative positions to capture an appropriate interdependency.

Starting from the initial estimation of the cross-section of ice clouds, the 2D OEM algorithm begins to iteratively minimize the cost function defined in Eq.(5). Similar to the 1D OEM algorithm discussed in Section 4.1, the 2D OEM algorithm employs the LM optimization method to iteratively adjust the state vector. The Jacobian matrix that captures the TB sensitivity to changes in each state element is also computed using the finite difference method. After perturbing a cloud quantity at a specific grid point, only the TB observations that interact with this voxel as identified by the ray-tracing step in section 2.2 are recalculated. Despite this effort, computing a 2D Jacobian matrix based on perturbation still requires substantial radiative transfer calculations. Additionally, the 2D OEM optimization requires significant memory resources because of the large 2D Jacobian matrix and the intensive matrix operations involved.

To manage computational constraints, we divide the entire set of scanning TB observations in Figure 2 into smaller sectors and process the cloud reconstruction for each sector independently. As noted in Section 2.1, the scanning radiometer completes 1,173 slices over the reference cloud scene, with each slice consisting of 97 pencil beams. We group each set of 51 scan slices into one reconstruction sector, and a total of 23 independent reconstruction sectors are created. Within each sector, approximately 200,000 radiative transfer calculations are needed to compute a single 2D Jacobian matrix. Due to the substantial computational burden, the number of iterations for the 2D OEM optimization is restricted to 9 in this study.

 **5    Simulation experiment results**

We conduct a simulation experiment to evaluate the performance of two tomographic reconstruction algorithms. After adding noise, the along-track scanning TB observations shown in Figure 4 are fed into each tomographic cloud reconstruction algorithm, and the reconstructed cloud microphysical parameters are compared to the true values. The added TB noise is assumed to be Gaussian, with the standard deviation set according to the NEDT specified in Table 1. The same radiative transfer model used to simulate observations and build the retrieval database is applied in the cloud reconstruction process. The intention is to eliminate the forward model error and assess the tomographic algorithms' performance in an idealized environment. To assess the advantage of using multi-angle observations, the tomographic reconstructed cloud results are also compared to the cloud profiles retrieved using nadir observations shown in Figure 3. The nadir-only retrievals are performed using the 1D hybrid BMCI and OEM algorithm presented in Section 4.1.

Figure 7 presents a direct comparison between the 2D reference cloud cross-section and those retrieved using different methods. The top two panels display the 2D reference IWC quantities alongside the nadir-only retrieval results, and the bottom two panels show the reconstructed 2D IWC structure based on the 1D and 2D tomographic reconstruction algorithms (referred to hereafter as Tomo-1D and Tomo-2D, respectively). The most notable characteristic of the nadir retrievals is the overall noisiness of the results. While the general cloud structure is captured, the transition of IWC profiles along the latitudinal range is not smooth. Instead, the retrieved IWC values show marked inconsistencies with abrupt transitions and fluctuations. For the dense cloud area around latitude $48°$, the retrieval results detect and portray the cloud formation, indicating that the TB observations contain sensitivities to large IWC particles. In contrast, in regions characterized by low IWC clouds such as the cloud top, the results become increasingly noisy and lack continuous cloud structure.

The most notable feature of the Tomo-1D reconstructed results is that the results become much smoother. Fewer abrupt changes and more gradual transitions of IWC across the latitudinal range are observed, and the depiction of the cloud structure is more refined. More spatial correlations and interactions between cloud particles along both vertical and horizontal dimensions are observed. The smoothness and refinement improve the overall readability of the results and make it easier to identify trends and patterns in the cloud spatial distribution. However, the Tomo-1D method exhibits limitations in improving spatial resolution, as limited cloud details are revealed whether at the cloud top or bottom.

The Tomo-2D algorithm demonstrates the best performance in reconstructing the 2D cross-section of ice clouds. It effectively captures the cloud structure and reconstructs the inner cloud details with high spatial resolution. The reconstruction capability is most evident at the cloud top and in areas with high IWC. The reference clouds show some extreme volumes around 8 km in altitude, and the Tomo-2D algorithm provided a detailed reconstruction of these features. In dense cloud regions such as the left side of latitude $46°$, the reconstructed cloud closely matches the reference pattern. The reconstruction capability at the cloud bottom shows some limitations. For instance, in the bottom layers between latitudes $47°$ and $50°$, the reconstructed results reveal two visible cloud bulks, but finer details are missing. Also, the results show signs of noise at the lowest cloud layers.

Figure 8 shows the TB residuals as a function of latitude and scanning angle to assess the convergence of the Tomo-2D algorithm. The TB residual is defined as the difference between the simulated and observed brightness temperatures ($TB_{sim}$-$TB_{obs}$). The left panel shows the TB residuals for the initial 2D cloud assumptions where the 2D OEM optimization starts, and the right panels show the TB residuals after the 2D OEM optimization has been applied. Note that due to computational constraints, the number of OEM iterations is limited to 9. The color bar range for the left panels is set to ±10K, and it is narrowed to ±2K in the right panels to improve the clarity of the results. The results show that the TB residuals decrease significantly after applying the 2D OEM optimization procedure. In the left panels, large discrepancies are observed across most scanning views. In certain regions such as around latitudes 45° and 47°, the TB discrepancies for some channels exceed 10K. In contrast, the right panels display a substantial reduction in discrepancies as most colors vanish. The TB simulations of 183 GHz channels in the top four panels show good agreement with the observations, and the TB discrepancies fall within the measurement uncertainties. The 325.15±11.5 GHz and the 684 GHz channels still show some discrepancies, with the simulations generally being warmer than the observations. As shown in Figure 2, these two channels are the most sensitive to IWC variations. The TB residuals demonstrate the effectiveness of the 2D OEM optimization in gradually converging to a stable solution using the a priori constraints derived from the Bayesian MCI results.

Figure 9 shows the 2D histogram comparing the retrieved IWC values against the true values. The nadir-only retrieval results and the tomographic reconstructed results based on the Tomo-1D and Tomo-2D algorithms are shown respectively. The solid line represents the median values of the retrieved IWC within each specific true IWC range, while the dashed lines indicate the corresponding interquartile range (IQR). The nadir-only retrievals exhibit the most scatter and deviation from the ideal 1:1 diagonal, with the retrieval data spreading across approximately two orders of magnitude throughout the entire IWC range. The nadir-only retrievals show a tendency to underestimate true values at the high end and overestimate them at the low end. In the middle range, the median line aligns with the diagonal, and no systematic biases are observed. The non-biased feature is mostly due to the identical radiative transfer model used for both simulating observations and performing the retrievals. The IQR remains constant at the higher IWC range and starts to spread below approximately $10^{-5}\ kg\ m^{-3}$, indicating the detection limitation for small IWC particles.

The Tomo-1D reconstructed results show a tighter clustering of points. The median line remains unchanged, indicating a limited capability in correcting the systematic biases at both high and low ends. The IQR extending to the low end becomes noticeably narrower. The Tomo-2D results in the third panel show the tightest clustering around the diagonal. At the high end, the underestimation is significantly corrected, with the median line closely following the diagonal to the very top end. The overestimation bias at the low end is improved. The improvements in the IQR are substantial, with the IQR narrowing significantly over the entire IWC range. The best performance is observed when IWC is around $10^{-4}\ kg\ m^{-3}$, where the IQR is the narrowest. Even at the low end with IWC values below $10^{-5}\ kg\ m^{-3}$, the reduction in IQR is evident, suggesting that the tomographic reconstruction technique effectively increases the detection sensitivity for small ice cloud particles.

Figure 10 shows the box plot of the logarithmic error in dB as a function of IWC for different retrieval methods. The log error is defined as:

$$log_{err} = 10 * log_{10}(\frac{x_{ret}}{x_{true}}) \tag{9}$$

where $x_{ret}$ and $x_{true}$ represent the retrieved and true parameters, respectively. The box plot provides a quantitative summary of the scatter plot in Figure 9. For nadir-only retrievals, the retrieval errors are large when the IWC drops below $10^{-5}$ $kg$ $m^{-3}$,
with whiskers extending beyond $\pm 10$ dB. As the IWC increases to the $10^{-5}$ to $10^{-4}$ $kg$ $m^{-3}$ range, the errors stabilize, with the whiskers narrowing to a spread of -5 to 5 dB and the IQR of around 4 dB. The most accurate retrievals occur when the IWC values are around $4 * 10^{-4}$ $kg$ $m^{-3}$. At higher IWC values, the results start showing biases. The Tomo-1D method does not show the ability to correct the median bias, but it demonstrates a modest reduction in both the whisker spread and the IQR. The Tomo-2D method evidently corrects the biases at both high and low IWC ends and significantly reduces the deviation of
retrieval errors over the entire IWC range. The reconstructed parameters are highly accurate in the $10^{-4}$ and $10^{-3}$ $kg$ $m^{-3}$ IWC range, with the IQR reduced to approximately 1dB.

Figure 11 shows the box plot of the logarithmic error in dB at different altitudes for the three retrieval methods. Only the voxels with the reference IWC values larger than $10^{-6}$ $kg$ $m^{-3}$ are used in this analysis. In the nadir retrievals, there is a noticeable trend of underestimation at high altitudes and overestimation at lower altitudes. The IQR remains nearly constant
at higher altitudes, and the narrowest IQR observed is at approximately 4 km, corresponding to the altitudes with the densest cloud coverage. Similar to Figure 10, the Tomo-1D method demonstrates the ability to reduce the whisker spread and IQR, but it is unable to correct the systematic retrieval biases. The Tomo-2D method shows marked improvements in both correcting biases and reducing retrieval error deviations. The median errors are very close to zero below 10 km in altitude, and the deviation of logarithmic errors narrows to around 1dB. Clear improvements are observed down to the cloud bottom, indicating
the penetration capability of the CoSSIR channels and their effectiveness in reconstructing the internal structure of clouds.

## 6 Summary and conclusions

This paper presents our latest development in tomographic cloud reconstruction algorithms that use multi-angle TB observations to reconstruct the spatial distribution of ice clouds. The developed techniques advance the submillimeter-wave radiometer's capability in retrieving ice cloud vertical profiles and contribute to enriching the theoretical foundation of the tomographic
reconstruction methods in the remote sensing field.

Two types of tomographic reconstruction algorithms have been developed. Both algorithms are within the Bayesian framework and use a hybrid Bayesian MCI and OEM optimization approach. The first algorithm performs 1D retrievals for each TB beam at a specific position and viewing angle. Using the geodetic coordinates of the voxels on the ray path provided by the ray-tracing technique, this algorithm conducts 1D retrievals beam by beam to determine cloud parameters along the propaga-
375 tion path. The MCI is initially applied, and if it fails due to insufficient database cases, the 1D OEM is employed to further minimize the cost function. All 1D retrieved parameters and uncertainty estimates expressed in a Gaussian form are averaged to

produce the final 2D cloud microphysical parameters. The second algorithm employs a more straightforward approach, which uses the scanning TB observations to simultaneously constrain the 2D cloud microphysics. An initial 2D cloud assumption is constructed using the 1D Bayesian MCI retrieval results for each radiation beam. With an appropriate prior constraint, the 2D OEM algorithm employs LM optimization to iteratively refine the 2D cloud microphysics and reduce the cost function. The Jacobian matrix is computed using the finite difference method to provide the necessary gradient information.

A simulation experiment is conducted using simulated observations over a reference cloud cross-section obtained from the CloudSat product. The CoSSIR radiometer operating on along-track scanning mode is positioned on an imaginary ER-2 aircraft following a trajectory identical to CloudSat's orbit. The scanning TB observations are simulated by splitting them into a series of 1D pencil-beam radiations based on the IBA simplification. A multi-angle retrieval database is established using the CloudSat product to obtain a priori information. The simulated observations are fed into the tomographic reconstruction algorithms, and the reconstructed 2D IWC profiles are compared against the true values to assess the tomography algorithms' performance. Additionally, these reconstructed clouds are compared to the results retrieved from nadir-only observations to evaluate the relative improvements achieved through multi-angle observations.

The simulation experiment demonstrates stable convergence of both tomographic reconstruction algorithms and highlights the significant advantages of using multi-angle observations to constrain 2D ice cloud parameters. In the nadir-only retrievals, although the TB observations show some sensitivity to large IWC voxels, its capability to reproduce the vertical profile of ice clouds is very limited. The constructed cross-section of ice clouds lacks structural continuity and features abrupt transitions between independent IWC profiles. Also, the retrieved parameters show systematical biases and considerable deviations in retrieval errors. Using multi-angle TB observations significantly improves the ice cloud reconstruction performance. Two types of tomographic reconstruction algorithms show different performances, with each having its own advantages and disadvantages.

The Tomo-1D tomographic algorithm shows clear improvements in smoothing cloud structure. The noise between adjacent vertical profiles is reduced, and the inner cloud structure becomes more distinct. However, the improvement is limited as the reconstructed cloud cross-section does not have high spatial resolution. While the Tomo-1D algorithm effectively reduces the deviation of retrieval errors, it is unable to correct the systematic biases. The advantage of the Tomo-1D algorithm is its ease of implementation and minimal demand for computational resources such as memory. The Tomo-2D algorithm exhibits the best reconstruction performance. The cloud structure is accurately captured, and the inner details of the cloud are well reproduced with high resolution. The Tomo-2D method effectively corrects systematic biases and reduces the deviation of retrieval errors. The retrieval accuracy is improved across a wide range of IWC values and altitudes. Also, the Tomo-2D algorithm increases the detection sensitivity for small ice cloud particles. The main disadvantage of the Tomo-2D method is the complexity of its coding and the significant computational resource requirements, both of which pose challenges to its implementation.

Several limitations of the current algorithms and areas for future improvement are worth mentioning. The biggest bottleneck in the current Tomo-2D algorithm is the calculation of the 2D Jacobian matrix. The current Jacobian calculation is based on the finite difference method, which is accurate but computationally extremely expensive. Exploring new methods to accelerate Jacobian calculation will be a key focus moving forward. Also, the performance of the tomographic algorithms in more complex experimental settings such as variations in PSD and particle shapes, as well as in more sophisticated atmospheric conditions

containing supercooled liquid water, remains to be evaluated. Additionally, this study employs the pencil-beam assumption that treats each beam as an infinitesimally narrow ray, and the impact of the radiometer's real-world beam width and footprint on the reconstruction accuracies has not been investigated. Last but not least, the current tomography algorithm does not utilize polarization information, which is particularly valuable at large viewing angles. Incorporating polarized observations into the reconstruction algorithm is expected to further improve retrieval performances. In the next study, the updated version of the tomographic cloud reconstruction algorithm will be applied to actual measurements from the airborne IMPACTS field campaign to further advance its development.

*Data availability.* The CloudSat data used to evaluate the performance of tomographic reconstruction algorithms can be accessed publicly on the CloudSat Data Processing Center website: https://www.cloudsat.cira.colostate.edu/.

*Author contributions.* Yuli Liu developed the tomographic reconstruction algorithms, conducted the simulation experiments, and wrote the manuscript. Ian S. Adams, as the principal investigator of this project, provided guidance on the methodology development and results assessment, as well as revisions to the manuscript.

*Competing interests.* The authors declare that no competing interests are present.

*Acknowledgements.* The authors would like to thank Dr. Mircea Grecu for providing comments and guidance on the manuscript. The authors also express their gratitude to the CloudSat community for their dedication in providing data products at different levels. Additionally, the authors extend their appreciation to the ARTS community for their continuous efforts in advancing the radiative transfer model.

*Financial support.* This work was funded by NASA's Airborne Instrument Technology Transition (AITT; NNH19ZDA001N-AITT), and additional funding was provided through the IMPACTS field campaign by the NASA Earth Science Division (ESD) and Earth Venture Suborbital Program under the NASA Airborne Science Program.

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

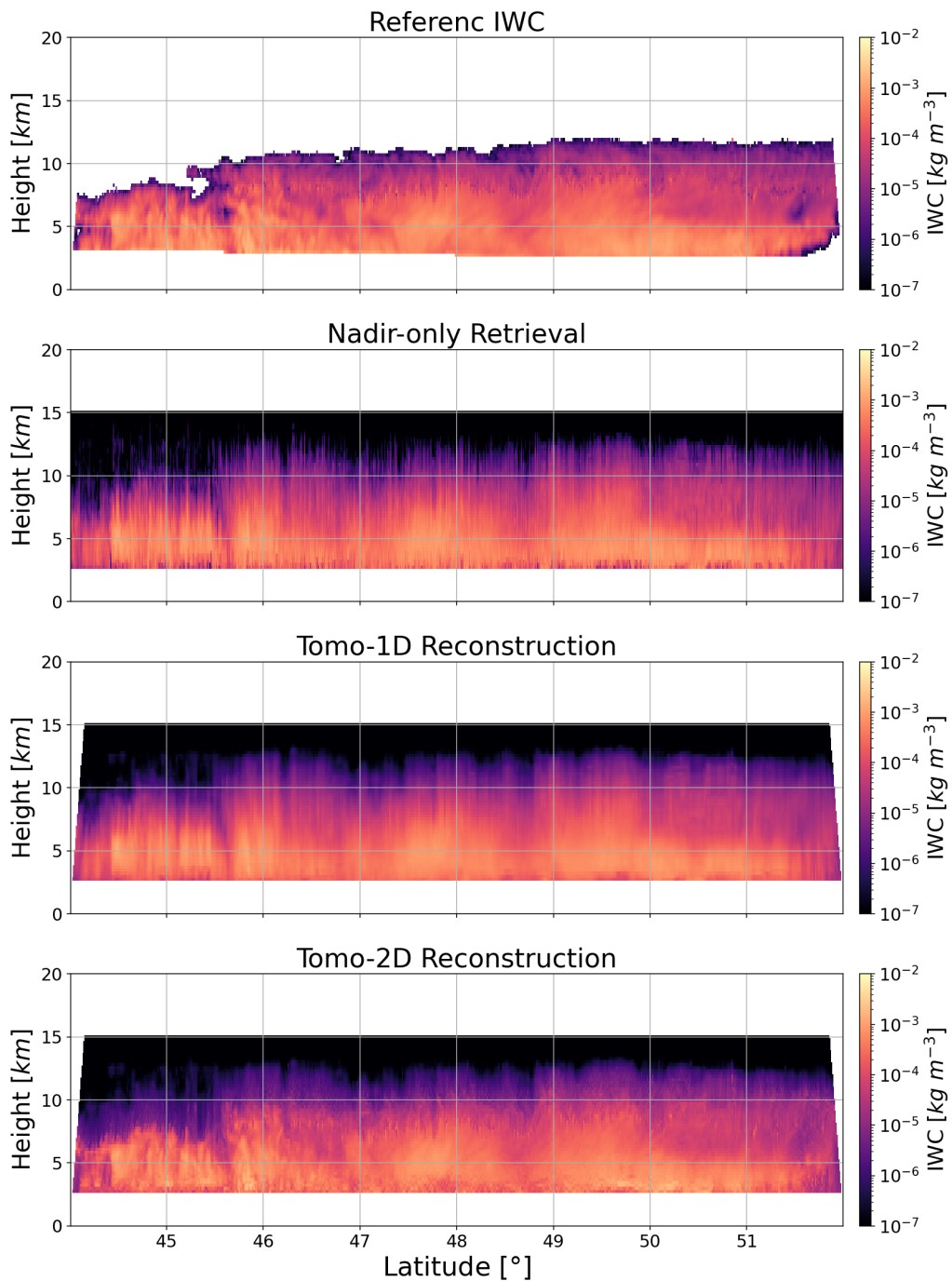

**Figure 7.** The top two panels show the reference IWC cross-section alongside the retrieval results obtained using nadir-only observations, and the bottom two panels show the reconstructed 2D IWC quantities based on the 1D and 2D tomographic reconstruction algorithms (referred to as Tomo-1D and Tomo-2D, respectively).

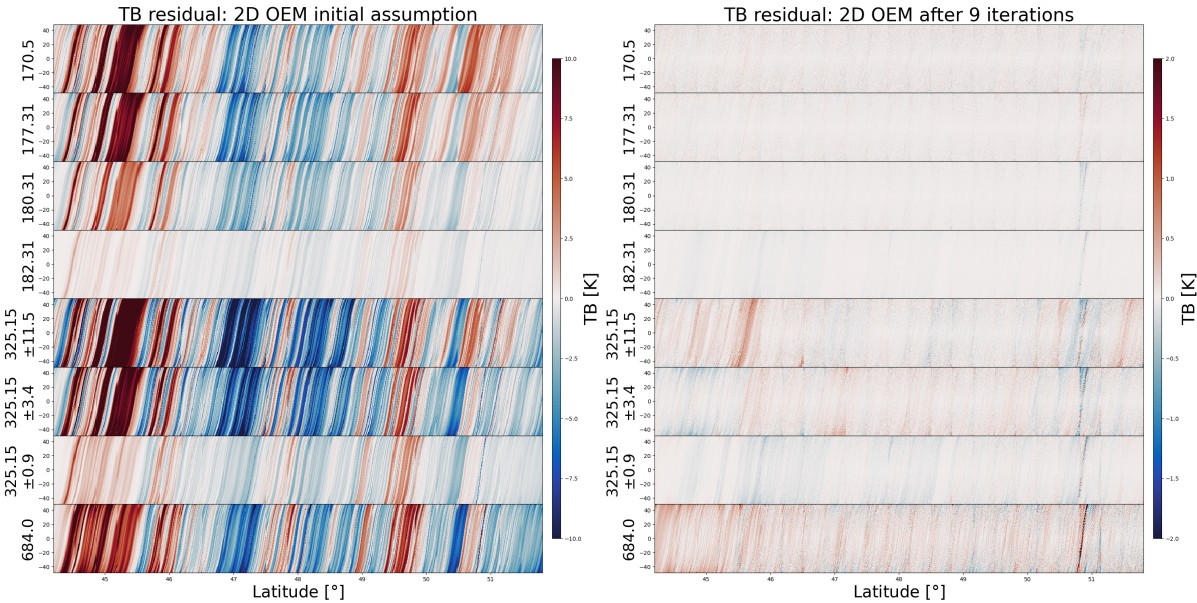

**Figure 8.** TB residuals for different CoSSIR channels as a function of latitude and scanning angle before and after applying the two-dimensional OEM optimization algorithm. The OEM iteration is limited to 9 due to computational constraints.

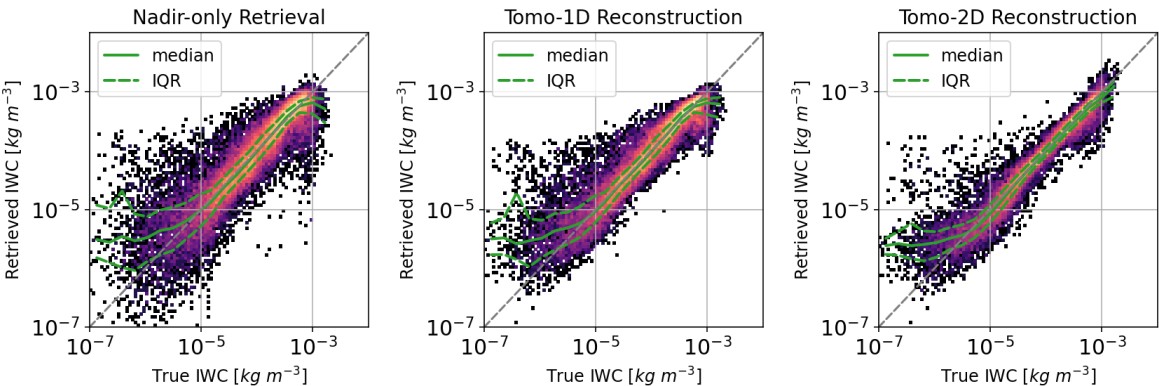

**Figure 9.** The 2D histogram compares the retrieved IWC values with the true values for the retrieval results using only nadir observations, as well as the tomographic reconstructed results based on the one-dimensional (Tomo-1D) and two-dimensional (Tomo-2D) tomographic algorithms, respectively.

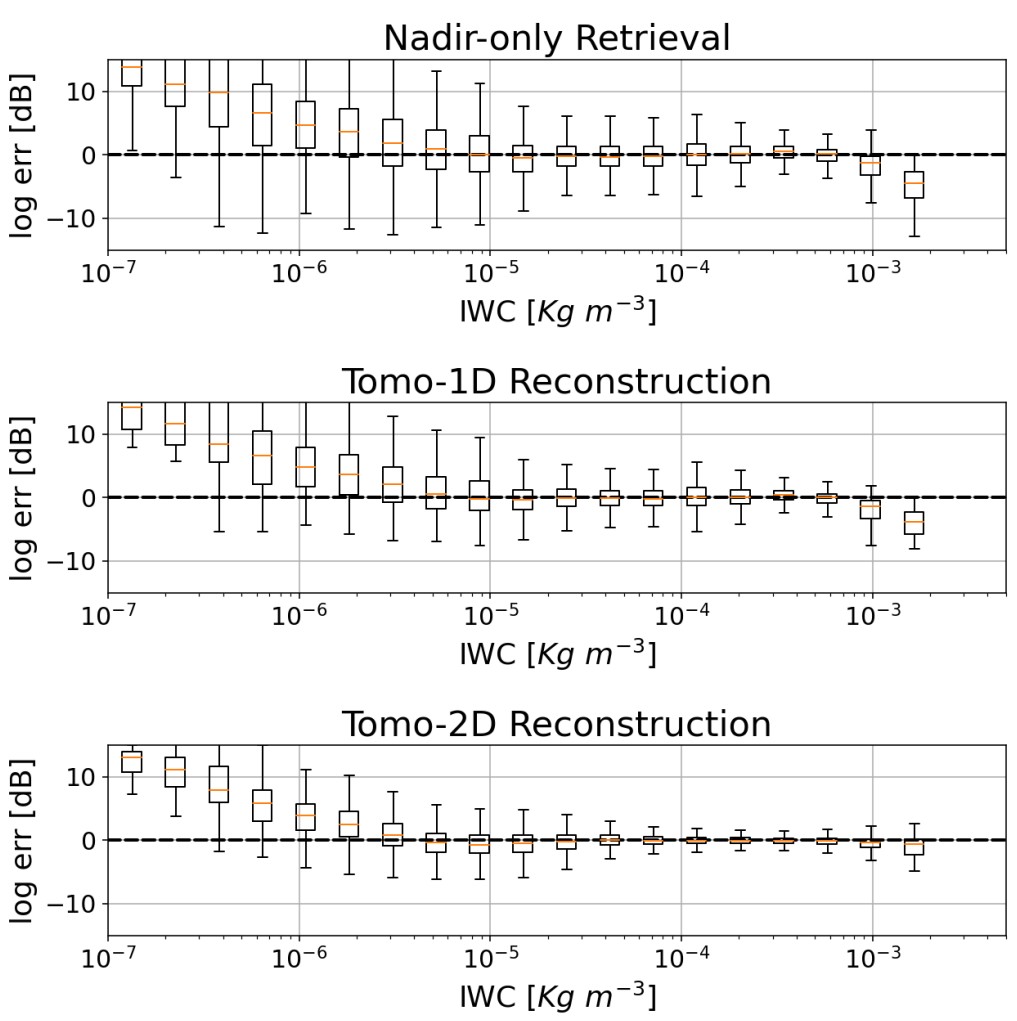

**Figure 10.** The box plot of the logarithmic error in dB as a function of ice water content (IWC) for the retrieved results using nadir observations, and the reconstructed results using the one-dimensional (Tomo-1D) and two-dimensional (Tomo-2D) tomographic reconstruction algorithms, respectively.

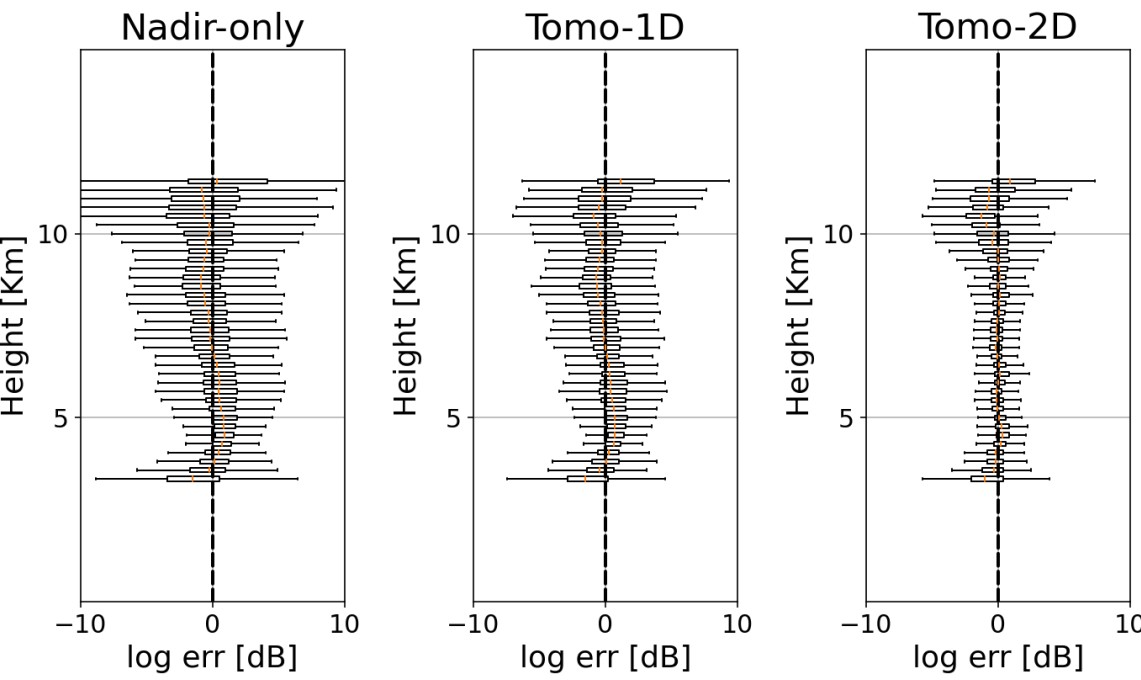

**Figure 11.** Same as Figure 10 but for the box plot of the logarithmic error as a function of altitude.