# Peer review of "Tomographic reconstruction algorithms for retrieving two-dimensional ice cloud microphysical parameters using along-track (sub)millimeter-wave radiometer observations"

_Atmospheric Measurement Techniques, 2024_

## Author Comment (AC1)

**Responses to Comments**

We sincerely thank Editor Dr. Xu, and two anonymous reviewers for the constructive and thoughtful comments.
Comments are in blue italic lettering, responses in black.

**Reviewer 1 Comments**

Line 79. Is the current CoSSIR not capable of along-track scanning?

CoSSIR, and its sibling sensor CoSMIR, have two-axis gimbals that allow programmable scan patterns. One axis rotates the scan head, which contains the receiver electronics, in elevation, while the structure holding the scan head rotates in azimuth. This entire structure is called the scan pedestal, and it enables heritage scanning modes of conical and cross track, as well as hybrid combination of modes. Since both sensors observe directly through the feeds mounted on the scan head, the instrument scans with a fixed polarization orientation, i.e., the polarization does not rotate when performing a cross-track scan. The along-track mode is simply the cross-track scanning mode at 90º azimuth.

During the first deployment of IMPACTS in 2019/2020, our team tested a hybrid along-track/conical mode in anticipation of future CoSSIR flights. CoSMIR was then operated in this mode for the entirety of the 2021/2022 deployment. Finally, the updated CoSSIR with the channel set listed in this manuscript flew in the 2022/2023 deployment, operating in the hybrid along-track/conical model through the entire deployment.

CoSSIR has not been operated solely in along-track mode, but CoSMIR was recently operated in this mode during the West-Coast and Heartlands Hyperspectral Microwave Intensive Experiment (WH$^2$yMSIE) for the first (test) flight.

Line 99. Since CloudSat is not scanning, meaning that an orbit of 2C-ICE is striped, how to understand the 3D scene here?

A 2D simulation framework is sufficient for the along-track simulations with pencil-beam assumptions in this study. The purpose of using a 3D framework is to allow for future extensions to include conical and cross-track scanning, as well as real-world observations that require consideration of factors such as beam width and airplane rotation. More descriptions have been added, as shown in Lines 109-111 in the track-changes document.

Line 101. How to understand the IBA technique here, whether it is by averaging 1D simulated observations within the field of view or by extracting multiple atmospheric profiles at different locations on the path to synthesize a scene?

The IBA is the underlying assumption behind the ray-tracing method. It first uses ray tracing to determine the propagation path and extract the atmospheric/cloud parameters along that path. Then, it applies the 1D forward model to simulate the corresponding TB. The description of IBA has been modified, as shown in

Lines 102-104 in the track-changes document. The ray-tracking method with IBA is described in detail in Lines 114-117.

Figure 5. What spatial extent of cloud parameters are reconstructed in one along-track scan, is it the entire scan? How do you handle the front and back scans where there is a crossover?

The cloud reconstruction step uses the 1D retrieval results from all 1,173 scans. The reconstruction process is carried out from the perspective of the grid points in the 2D cross-section. For a specific grid point, if there are pencil beams interacting with it, the cloud parameters at that point are constructed by averaging the corresponding 1D retrieval results. In cases where consecutive scans overlap, the TB observations from both sides contributed to the cloud reconstruction at the crossover grid points. The figure below shows a small section of the entire along-track scans as an example. The 2D cross-section of grid points are colored by the number of pencil-beams interacting with each point. This visualization helps to illustrate how the pencil beams interact with cloud parameters at different grid points. The interactions between the pencil beams and grid points are determined during the ray-tracing step with IBA.

[Figure]

Line 190. If the BMCI performs successfully, does the OEM not execute, why the minimum database case is 25?

Line 197. Does the local Gaussian a priori constrain mean the a prior IWC and covariance matrix used for OEM?

Line 197. That is, even if the BMCI fails, at the start of the OEM, it still has to find 25 cases in the database to update the a posteriori PDF. I'm a little confused here.

Line 237. How to understand auto-adjustable BMCI? Is it any different than the BMCI in 1D?

These four comments are related to the BMCI's step to automatically adjust the measurement uncertainty, and they are addressed together here.

The issue with the original BMCI arises when there are insufficient database cases that match the input TB within the measurement uncertainty range. In such cases, if the original BMCI is applied, only a small number of database cases are assigned significant weights, while the majority receive weights close to zero. This causes the weighted sum to be overly influenced by just a few cases, resulting in a computed standard deviation that becomes unreasonably small.

To address this issue, we can inflate the measurement uncertainty to expand the search range and include more database cases within the threshold. This approach ensures a sufficient diversity of database cases with meaningful weights, which leads to a more reasonable estimate of retrieval uncertainty. This

approach was first proposed by Evans et al. (2005) and has demonstrated its strength in our previous studies.

The produced uncertainty estimates are used as prior constraints during the OEM. The setup of Sa is described in Lines 216-217 in the track-changes document. After inflating the measurement uncertainty, the resulting BMCI uncertainty becomes correspondingly larger. This indicates that, due to the finite number of database cases, we are unable to find enough cases similar to the given TB. As a result, we loosen the prior constraint during the OEM in a proportional manner. Consequently, the OEM will place more emphasis on the agreement between the simulated and observed TB.

More descriptions of BMCI have been added, as shown in Lines 209-211 in the track-changes document.

Line 265. Computational resources and speed are really an issue with this algorithm, roughly how much computational resources and computation time are needed at the moment?

The 2D tomographic algorithm is currently very computationally expensive. As mentioned in Lines 274-280 in the track-changes document, the reconstruction curtain includes 23 sectors, and for each sector, approximately 200,000 DISORT runs are required for a single Jacobian calculation. With 9 iterations performed in this study, this results in a total of $200{,}000 \times 9 \times 23$ DISORT runs using the perturbation Jacobian method. We are now exploring new methods to accelerate the Jacobian calculations. One approach we've tested is a semi-analytical method, which reduces the computational cost by approximately two orders of magnitude. Preliminary results show that the computed Jacobian still allow the OEM to converge to a stable state. Additionally, we are investigating the use of a neural network (NN) approach, which, if successful, could provide very fast Jacobian calculations. We will present further advancements in future studies.

---

## Author Comment (AC2)

**Responses to Comments**

We sincerely thank Editor Dr. Xu, and two anonymous reviewers for the constructive and thoughtful comments.
Comments are in blue italic lettering, responses in black.

**Reviewer 2 Comments**

*Line ~110 – The results presented here depend to some extent on the details of the particles that were assumed to make up the liquid and ice clouds, and how supercooled water was handled. To make the results more reproducible, I would encourage the authors to make their PSD assumptions explicit. Because the profiles are artificially simple, the authors should also add a caveat to the conclusions. It may be that changes in PSD properties or the inclusion of super-cooled water in the cloud may be another reason to perform 2D tomography but that has not been demonstrated.*

The PSD for liquid clouds has been added along with the PSD for ice clouds, as shown in Lines 122-123 in the track-changes document. The particle habit assumption for both ice and liquid particles are also described in Lines 121-122. Additionally, in the conclusion section, the limitations of this study in terms of simplified experiment setup and atmospheric conditions have been discussed, as shown in Lines 411-413,

*Line ~140 – the suggestion is that Tb are computed for each CloudSat profile at multiple angles needs a little more explanation. Because slant path computations cut through multiple horizontally adjacent profiles, this leads to uneven layering of the slant path profiles. Exactly how the authors handled this was not clear from the description.*

*Line ~150 – If Tbs are constructed from single profiles but simply for different view angles, then the vertical correlations are assumed in the prior data. Some discussion here about how this is ultimately handled in the retrieval would be appropriate here.*

Both comments concern the prior database and are addressed together. The current database is built by applying multiple angles to the same vertical profile from the CloudSat product. Addressing the actual slanting angle is very important, but this task has not been done yet. This simplification reduces the variability of cloud parameters in the prior database and affects the correlation between clouds at different layers. The establishment of a more sophisticated prior database will be done next. The limitation of the current prior database has been described in Lines 154-158 in the track-changes document.

*Line 220 – Simply averaging the same voxels retrieved from each view angle in the 1-D scheme seems to artificially simplistic. Given that the authors have the goodness of fit from the OE, should this not be a weighted average?*

Thanks for the insightful suggestions. The current averaging method assumes that each TB observation from different angles contributes equally to the final results. As you suggested, the contributions can be weighted based on factors such as the retrieval uncertainty in the covariance matrix and how well the

simulations reproduce the TB observation indicated by $\chi^2$. More description of the weighted averaging method has been added in Lines 237-239 in the track-changes document.